# Implementation of a Sponge-Based Flexible Electronic Skin for Safe Human–Robot Interaction

**DOI:** 10.3390/mi13081344

**Published:** 2022-08-19

**Authors:** Kun Yang, Xinkai Xia, Fan Zhang, Huanzhou Ma, Shengbo Sang, Qiang Zhang, Jianlong Ji

**Affiliations:** 1Shanxi Key Laboratory of Micro Nano Sensor & Artificial Intelligence Perception, College of Information and Computer, Taiyuan University of Technology, Taiyuan 030024, China; 2Key Lab of Advanced Transducers and Intelligent Control System of the Ministry of Education, Taiyuan University of Technology, Taiyuan 030024, China; 3Shanxi Institute of 6D Artificial Intelligence Biomedical Science, Taiyuan 030031, China

**Keywords:** flexible sensor, human–robot interaction, tactile acquisition system, FDM 3D printing

## Abstract

In current industrial production, robots have increasingly been taking the place of manual workers. With the improvements in production efficiency, accidents that involve operators occur frequently. In this study, a flexible sensor system was designed to promote the security performance of a collaborative robot. The flexible sensors, which was made by adsorbing graphene into a sponge, could accurately convert the pressure on a contact surface into a numerical signal. Ecoflex was selected as the substrate material for our sensing array so as to enable the sensors to better adapt to the sensing application scenario of the robot arm. A 3D printing mold was used to prepare the flexible substrate of the sensors, which made the positioning of each part within the sensors more accurate and ensured the unity of the sensing array. The sensing unit showed a correspondence between the input force and the output resistance that was in the range of 0–5 N. Our stability and reproducibility experiments indicated that the sensors had a good stability. In addition, a tactile acquisition system was designed to sample the tactile data from the sensor array. Our interaction experiment results showed that the proposed electronic skin could provide an efficient approach for secure human–robot interaction.

## 1. Introduction

Throughout the process of industrialization, human beings have explored ways to liberate their hands and labor force to a greater extent. Since robots were first deployed in production lines in the 1960s, industrial production efficiency has improved greatly [1]. However, due to the large inertia and high speed of traditional robots, accidents are common, especially when robots cooperate with human operators. Frequent close contact with robots can cause potential threats to the personal safety of operators or even direct injuries [2]. In 2016, an operator at an auto parts manufacturer in Alabama was crushed and killed by a robot. In December 2019, an accident involving a robot occurred in Amazon’s automated warehouse, which resulted in 24 employees being injured and rushed to hospital. In production processes, accidents that are caused by robots occur frequently, which not only results in a waste of production resources but also places heavy physical and mental burdens on operators. In the long run, accidents that involve robots could greatly hinder the production processes of enterprises.

At present, passive security methods, such as electronic fences or industry standards, are usually used to ensure the security of workplaces in the manufacturing industry, but these passive security methods have some limitations, such as poor flexibility and real-time security. Therefore, finding ways to improve the active safety performance of operators has become a new research topic [3].

Safety standards that are related to collaborative robots have been proposed to prevent human–robot collisions and reduce the risks to operators [4], such as ISO 10218 and ISO/TS 15066:2016. A Canadian enterprise called KINOVA launched a Jaco assistive robotic arm to assist people with limited or no upper limb mobility to carry out movements that were previously impossible for them [5]. However, due to limited sensors, the Jaco arm cannot completely avoid collisions during these assistive operations. Hence, the realization of inherently safe interactions between humans and robots is still a challenge, both in industrial production and daily life [6].

In order to address the above challenges, researchers have begun to study active safety detection methods, including sensorless detection, visual detection and tactile detection. The most widely used sensorless detection method is based on a generalized momentum-based disturbance observer. This method was first proposed by De Luca et al. [7] in 2003 and achieved good collision detection between balloons and humans and a robotic arm [8,9]. However, this method cannot achieve precise position detection and has poor collision calculation capabilities. Visual detection methods have developed rapidly over recent years, as have the corresponding image processing algorithms that are also used to achieve collision prediction or detection [10,11,12,13,14]; however, vision sensors fail easily in high-speed collision detection. Therefore, an accurate and fast detection method needs to be developed urgently.

With the birth of the concept of flexible electronics, researchers have started to study electronic skins from the perspective of bionics and use electronic skins to provide sensing capabilities for robot arms [15]. In 2004, the Takao Smeya team in Japan [16] developed an electronic skin that could simultaneously obtain contact force and temperature information. This electronic skin was used for pressure tests on a robot hand to realize grasping movements. In 2005, the Yamada team in Japan [17] designed a soft electronic skin that could accurately sense the location of objects that were in contact with it. Researchers have subsequently studied electronic skins with increasing interest. In order to enhance the safety of humans during human–robot collaboration, Pang et al. [18] proposed a new collaborative electronic skin (CoboSkin). This electronic skin was made from soft and porous materials, which created a type of flexible sensor with variable sensitivity. The reliability of this electronic skin was verified in robot collision experiments. Saadatzi et al. [19] presented a flexible tactile sensor array that could multimodal tactile feedback to perceive the immediate surroundings of a robot arm. These sensors were fabricated using gold electrodes and micro-patterned piezoresistive polymers (PEDOT:PSS), which could provide the robot arm with pressure detection capabilities. Yan et al. [20] presented a soft tactile sensor that had self-decoupling and super-resolution abilities. Their design ideas for the flexible sensor were inspired by the fact that human skin can sense subtle changes in both normal and shear forces. Ji et al. [21] designed a flexible tactile sensing array that was based on a capacitive mechanism. The flexible tactile sensing array was made from polydimethylsiloxane (PDMS) dielectrics and had a high sensitivity, which was in the force range of 0–1 N. Therefore, this sensing array could provide robots with the capabilities to detect small external forces; however, the stability and reproducibility of the sensing array was not verified. So as to replicate the human sense of touch, Weichart et al. [22] proposed a sensing technology that was manufactured from dense arrays of highly sensitive tactile sensors on a flexible substrate. The softness of this flexible substrate was close to that of human skin, so the sensor could be better attached to a robotic arm. They also developed a piece of electronic equipment that could collect the values from the sensor at a high frequency. Pang et al. [23] designed and fabricated a 3D flexible sensor that was made from a piezoresistive nanocomposite. The reliability and performance of this sensor was verified using a YuMi robot. In order to obtain external information and ensure the security of human–robot interaction, Wu et al. [24] proposed a tactile sensor that had a double sensitive layer structure. The sensor converted information from external collisions or contact into local conductivity changes, based on the EIT (electrical impedance tomography) method. Shi et al. [25] designed and fabricated a novel flexible capacitive pressure sensor that had a micro-structured composite dielectric layer (MCDL). This flexible capacitive pressure sensor had quite a broad linearity detection range and a fast response time of 150 ms. To ensure the safety of humans in human–robot interaction tasks, Teyssier et al. [26] proposed a novel approach to designing and fabricating compliant human-like artificial skin sensors for robots. These sensors had similar mechanical properties to human skin and were capable of precisely detecting touch due to the use of different silicone elastomers. Zhang et al. [27] proposed a novel flexible tactile sensor that could detect the positions of pressure and force simultaneously. This tactile sensor adopted a three-layer structure (a conductive film, piezoresistive film and aluminum foil) to achieve the detection of force values and positions. In order to address the remaining issues of the sensitivity and response time of electronic skins, Lü et al. [28] proposed a flexible electronic skin that had a high sensitivity and a fast response time, which was based on piezoresistive graphene films. These sensing units were prepared using graphene/polyethylene terephthalate films and the substrate that was used was polyimide. Compared to sensorless interactive detection and visual detection methods, electronic skins achieve higher sensitivity and faster response times. However, electronic skins still need to be improved in terms of stability and applicability.

In this study, a strategy for safe human–robot interaction was developed and tested using a KINOVA robot arm, as shown in Figure 1. To develop this strategy, a flexible electronic skin for the robot arm was designed to realize tactile perception. The electronic skin was made by adsorbing graphene into a sponge. The electronic skin could also act as a buffer between the robot and external collisions, which could reduce damage to humans and the robot arm.

The rest of this article is organized as follows. The fabrication processes of the electronic skin are described in Section 2. The characteristics of the electronic skin and collision tests are discussed in Section 3 and the design of the tactile acquisition system is also described in this section. Finally, in Section 4, our conclusions are presented.

## 2. Materials and Methods

### 2.1. Design of the Electronic Skin Structure

One of the key challenges in achieving tactile perception abilities for robots is how to measure the surface pressure of the robots accurately. This challenge is more demanding on the sensors. Therefore, it has become necessary to design stable and durable flexible tactile sensors that can measure pressure. Over recent years, high-performance flexible sensors that use flexible substrates have been widely reported, including polydimethylsiloxane (PDMS), polyimide (Pl), polyethylene terephthalate (PET) and Ecoflex. By directly coating or fusing high-conductive materials, such as carbon nanotube (CNT), graphene (GR), onto black and silver acetylene nanowires, conductive networks can be established on flexible materials to prepare flexible strain sensors. In order to improve the sensitivity of the sensors further, it has become common to upgrade the 2D conductive networks to 3D networks. As a common substance, sponges have natural and stable pentagonal internal network structures that are composed of formaldehyde–melamine resin. Due to its low cost, high porosity, ultra-light weight and robustness, sponges are perfect starting materials for the fabrication of various sensors. The adsorption properties of conductive materials is the key to preparing stable 3D sensing units. Graphene is a 2D carbon nanomaterial that has a hexagonal honeycomb lattice, which is composed of carbon atoms with hybrid orbitals, and good electrical conductivity. Because of its 2D structure, graphene is easily adsorbed onto organic matter. In this study, a graphene–sponge sensor array was designed to accurately convert contact surface pressure into numerical signals. The elastic modulus of Ecoflex is the closest to that of human skin, so Ecoflex was selected as the substrate material for our sensing array in order to enable the sensors to better adapt to the sensing application scenarios of the robot arm.

The structure of the flexible tactile sensor array is shown in Figure 2. From bottom to top are the Ecoflex flexible shell (substrate), the lower electrode layer, the sensing unit composite layer and the upper electrode layer. The sensing unit was prepared by adsorbing graphene onto a sponge. When pressure was exerted on the sensing unit, the sponge skeleton compressed and the graphene sheets that were adsorbed on the sponge skeleton became closer together. Under the influence of the tunnel effect, more conductive paths were formed and the resistance of the sensors showed a decreasing trend. When the pressure on the sensing unit was released, the sponge skeleton recovered, the conductive paths were broken and the resistance increased. In addition, the flexible shells of the sensors were also an important part of its function. A 3D printing mold was used to prepare the flexible shell (substrate) of the sensor, which made the positioning of each part within the sensors more accurate and ensured the unity of the sensing array.

### 2.2. Fabrication Processes of the Electronic Skin

According to the application requirements of this study, the higher requirements of the sensors for their longitudinal pressure sensing performance was one of the important reasons for the sandwich structure of our sensor array. The flexible substrate was made of Ecoflex material that had a thickness of 1 mm and the slot that was reserved for the sensing unit was 1 cm × 1 cm. The sensing unit spacing was set to 1 cm to ensure the effective detection area of the sensor array. The sensor array consisted of three parts: electrodes, the flexible substrate and sensitive cells. The assembly process is shown in Figure 3.

Step 1: The strain sensor mold was fabricated using an FDM (fusion deposition modeling) 3D printer. Polylactic acid was used as the printing material, which was extruded from a 0.4-mm printing nozzle at a processing temperature of 210 °C. The thickness of each layer of the mold was set to 100 μm and the filling density was set to 20% using the slicing software settings. Then, the fallback printing method was used and the polylactic acid was printed, as shown in Figure 4a. After the mold was printed, it was removed and put in alcohol to be cleaned.

Step 2: Ecoflex glue A and glue B were measured out at a ratio of 1:1 and stirred with a 2-mm mixer before being poured into the pre-prepared mold. The mold was then placed in a vacuum drying oven for 5-mm defoaming and transferred to an oven at 50 °C for 1 h. Finally, the Ecoflex film was carefully released from the mold to obtain the flexible substrate.

Step 3: Weighing paper was folded in half on an electronic balance and the graphene (1–3 layers; purity = 98%), which was purchased from which was purchased from Shenzhen National Technology Co. Ltd. (Shenzhen, China), was transferred onto the weighing paper using a medicine spoon. Precisely 0.2 mg of graphene was transferred into a 100-mL reagent bottle. Then, 40 mL of anhydrous ethanol was also transferred into the reagent vial using a pipette gun. The reagent bottle as a whole was placed in a cell crusher. The power was set to 50% and the waveform was set to a square wave. The graphene and alcohol mixture was ultrasonically dispersed for 20 min to obtain graphene dispersion.

Step 4: A melamine sponge, which was purchased from Shanghai Junhua New Material Co. Ltd. (Shanghai, China), was washed three times with deionized water and was then rinsed with alcohol. The washed sponge was dried at 70 °C in an oven for 1 h and then cut into several pieces of 1 × 1 × 0.4 cm^3^. After being immersed in the graphene dispersion for 5 min, the sponge blocks were transferred to the drying oven for 1 h. This process was repeated twice and the melamine sponge blocks changed from white to black. The sensing unit was then prepared.

Step 5: Conductive tape was cut into the desired shape and laid on the flexible substrate. Sixteen conductive units were placed on the lateral electrodes in their corresponding positions and then, the cut longitudinal electrodes were placed on the conductive units. Finally, the electrodes were encapsulated with the Ecoflex solution and each sensing unit was individually wrapped and protected with film tape.

The integrated tactile sensor array exhibited good flexibility, which enabled it to be attached to the KINOVA robot arm, as shown in Figure 4b. Figure 4a exhibits the flexible Ecoflex substrate.

## 3. Results and Discussion

### 3.1. Calibration of the Electronic Skin

Although each sensing unit in the flexible tactile sensor array was made using the same manufacturing process with the same structure, they were cut from a single piece of prepared sponge that was mixed with graphene, so the sensor characteristics of each unit could be slightly different. Therefore, each sensing unit needed to be calibrated and tested before actual use. The configuration of the testing platform was mainly composed of a tension and compression testing machine (ZQ-990B, Zhiqu Precision Instrument Co. Ltd., Dongguan, China), a source measurement unit (Keithley 2440, Tektronix, Beaverton, OR, USA) and a computer (PC). The ZQ-990B had two modes for position control and pressure control. The force resolution was 1/10,000 and the force measurement accuracy was less than 1%. The maximum load and measuring range were 2 kN and it was equipped with upper computer software, which could collect pressure and position data in real time. The Keithley 2440 had various measurement modes, such as position voltage measurement, current measurement and resistance measurement. It had a high measurement accuracy of 0.012%. The configuration of the testing system is shown in Figure 5.

The sensing unit was fixed on the surface of an immovable fixture that was integrated into the tension and compression testing machine. The output of the sensing unit was connected to the source measurement unit. To prevent the metal fixtures from affecting the resistance tests, insulation tape was pasted onto the surface of the metal fixtures. Because the movable fixture was too wide, more than one sensing unit was triggered during one press cycle. Therefore, a small iron rod was fastened to the surface of the movable fixture to ensure that only one sensing unit was pressed at a time. When the movable fixture moved downward, the iron rod moved it and generated a changeable contact force by pressing single sensing units. The contact force was equal to the force that was exerted by the testing machine. The force was recorded by the force sensors that were integrated into the testing machine. The raw contact force and resistance data were sent to the PC software for further data processing, storage and analysis.

The initial resistance of each unit was between 11.6 kΩ and 11.9 kΩ before the loading tests. The data from all of the sensing units were relatively close, so one of the units was selected as a benchmark unit. Figure 6a exhibits the data from the benchmark unit. This graph shows the one-to-one correspondence between the resistance output and the force input, in the range of 0–5 N. The resistance decreased when a force was applied to it. It could be seen that there was an approximately linear relationship ($R^{2}$ = 0.9653) between the resistance value of the sensing unit and the applied force. The function of the pressure–resistance relationship is marked in Figure 6a and could be expressed as follows:(1)$$R_{s}=-1.005F+11.608$$
where $R_{s}$ is the resistance of the sensing unit.

The results of the reproducibility and stability experiments are shown in Figure 6b. The initial resistance value of the sensing units was 11.2 kΩ. When an external force of 5 N was applied, the sensing unit resistance decreased to 3.6 kΩ. During the experiments, the loading range was 0–5 N, the loading speed was 300 mm/min and the test period was 800 cycles. The applied frequency of the load force was 0.08 Hz. The repeatability test curves demonstrate the good repeatability of the sensing units. After 800 cycles of testing, the performance of the sensors remained good and the resistance consistency of the sensors was good, especially when the load force was 0 N.

### 3.2. Design of the Tactile Acquisition System

In order to test the developed tactile sensor array, a real-time tactile sensing system was designed, which is shown in Figure 7. The raw pressure data from the sensing units were acquired by an acquisition circuit and transferred as voltage data to the control block, which was controlled by an STM32F103ZET6 chip (STMicroelectronics, Geneva, Switzerland). A 12-bit analog-to-digital converter (ADC) was integrated into the control block to acquire the voltage data. The mean filtering method was used to obtain the smooth data. The voltage data were finally transmitted to an ROS operation system on the computer and provided the basis for the control of the robot arm.

More details about the acquisition circuit are shown in Figure 8a. The acquisition circuit consisted of two analog switches (CD4051BE, Texas Instrument, Dallas, TX, USA), four divided resistors and an operational amplifier (LM324, Texas Instrument, Dallas, TX, USA). The variable resistor array in the figure represents the tactile sensor array. The array had four row channels and four column channels. The two analog switches were connected to both the row channels and column channels in the sensor array. The CD4051-1 switch connected the column channels and was used as the common input of the sensor array. The CD4051-2 switch connected the row channels and was used as the common output of the sensor array. The two analog switches selected the channels periodically, which meant that each sensor in the array was scanned in cycles. The operational amplifier constituted a voltage follower for buffering and isolating the output of the CD4051-2 switch. The output of the CD4051-2 switch was connected to the control block. The resistance of the divided resistors and sensors determined the final output of the circuit. The output of the CD4051-2 switch could be expressed as:(2)$$V_{out}=V_{cc}\frac{R_{d}}{R_{s}+R_{d}}$$
where $R_{d}$ is the resistance value of the divided resistors (5 kΩ) and $R_{s}$ is the resistance value of the sensor unit that was being scanned.

After the collecting the voltage values from the sensor units, the $V_{out}$ data were transmitted to the control block. After being processed by the A/D (64 Hz) and filter modules, the contact information was transferred to the computer (ROS) via a serial bus (4 Hz). The pressure values could be calculated using the filtered $V_{out}$ data. According to Equations (Equation 1) and (Equation 2) (the pressure–resistance relationship), the formula for the sensor pressure values could be expressed as:(3)$$F=\frac{1}{-1.005}\left(\frac{V_{cc}R_{d}}{V_{out}}-R_{d}-11.608\right)$$

Finally, the state of the robot arm was controlled by the ROS, according to the processed signals.

### 3.3. Response of the Tactile Sensor Array

In this part of the study, a finger touch test was carried out to verify the actual performance of the flexible tactile sensing system. The actual performance of the entire system was verified by collecting and displaying the pressure data from the sensing array in real time. The data were processed and displayed in MATLAB, as shown in Figure 9. The vertical axis in each picture indicates the force value of each sensing unit. The 3D histograms in Figure 9 show that the force values of the corresponding sensing units increased obviously when they were pressed. The rectangular array in the upper left corner of each sub-figure represents the pressing situation of the sensing array, the blue squares represent the pressed sensing units and the gray squares represent the sensing units that were not pressed.

The illustration in each sub-figure shows the positions of the finger touch tests. As can be seen from Figure 9a, the sensing units remained relatively stable when they were not being touched. Figure 9b–d indicate that the sensing units could identify the pressure distribution correctly when they were touched under different conditions. Although touching the sensor units could cause the other sensor units to fluctuate incorrectly, these fluctuations did not affect the correct identification of pressure distribution. From the finger touch experiments, it was verified that the sensing system could work normally and stably when a large surface area was being touched.

### 3.4. Human–Robot Interaction Experiments

In this part of the study, human–robot interaction experiments were carried out to test the reliability of the flexible tactile sensor system (Appendix A). In these experiments, the sensors were fixed onto a KINOVA GEN2 collaborative robot (J2N6S200). The ROS (robot operating system) was used to receive and process the sensor data and control the robot. Figure 10a shows the initial experimental setup (t = 0 s). At the beginning of the experiments, the robot arm was far away from the operator. Joint 1 moved at a constant joint speed (15 °/s) in a counterclockwise direction. The flexible tactile sensors were attached to the end of Link 2. During the movement of the robot, the operator touched the sensors with their hand.

In the human–robot interaction experiments, the response strategies of the robot arm to different pressure values from the flexible sensors were set. When the sensor pressure value was less than 1 N, the robot arm moved at its normal speed (15 °/s). When the pressure value was greater than 1 N but less than 5 N, the speed of the robot arm decreased continuously and with a deceleration of −12 °/s^2^. When the pressure value was greater than 5 N, the robot arm stopped moving immediately.

Figure 10a–d exhibit the whole experimental process. Within the first 2.14 s of the robot arm moving, the operator did not touch the flexible tactile sensors and the robot arm moved at a normal speed. Figure 10c shows when the operator touched the flexible sensors and pressed the sensors lightly (t = 2.14 s), but all of the pressure values from the sensors were less than 5 N, so the speed of Joint 1 decreased continuously. Figure 10c shows when the maximum pressure value from the sensors was greater than 5 N and the robot arm stopped moving (t = 2.65 s).

Joint 1 in the robot arm and the maximum force of the sensors are shown in Figure 11. It can be seen from the tow curves that when the pressure values from the sensors were different, the speed of Joint 1 was also different. Between 0 s and 2.14 s, the maximum force was less than 1N, so the robot arm moved at a normal speed. Between 2.14 s and 2.65 s, the maximum force was greater than 1N but less than 5 N, so the speed of Joint 1 decreased continuously. Between 2.65 s and 3.8 s, the maximum force was greater than 5 N, so the robot arm stopped moving immediately. The results of our human–robot interaction experiments proved that the flexible tactile sensor system could provide the robot arm with a good perception ability to deal with collisions.

### 3.5. Discussion

Although the sensing units were made from a single piece of sponge, the characteristics of each sensing unit could be slightly different due to errors in the preparation process. To verify the consistency of the units, a fitting analysis of the different sensing units was performed, which is shown in Figure 12.

The black points represent the average data from the remaining 15 sensing units under different pressures. The red line represents the fitting line, which was calculated in Section 3.1. Then, the $R^{2}$ value of the average data and the fitting line was calculated as 0.9849, which meant that the fitting line was a good representation of the averages. When the sensing units were pressed, the fitting line could be used to calculate the pressure values from the sensing units by calculating their resistance values.

The results from the calibration and stability tests of our tactile sensor units verified that the tactile sensors had good sensitivity and linearity. Furthermore, the sensors could maintain good performance under high-frequency pressing. These results proved that the sensors could be used in practical engineering applications to provide stable sensing capabilities for robot arms.

The sensors generally had two sensitivity intervals. In the early stages of applying pressure, large deformations of the 3D structures allowed the adsorbed conductive materials to form more pathways. Therefore, under lower pressures, the sensors exhibited high sensitivity. As the pressure increased, the sensor pathways were basically formed intact and the sensor sensitivity decreased. The two sensitivity intervals of the proposed sensors are illustrated in Figure 13. The ranges of the two sensitivity intervals were 0–5 N and 5–30 N. In order to detect collisions effectively, a high-sensitivity interval of 0–5 N was selected for this study.

For the purpose of increasing the sensitivity intervals of the sensors, the method of increasing the sponge thickness was adopted [29]. It is worth mentioning that the shape of the electrodes also affected the sensitivity intervals of the sensors [30]. The contact area of the electrodes and sponge was controlled by adjusting their shapes. When the pressure increased, the contact area increased and the contact resistance decreased. The resistance values from the sensing units and the contact resistance values acted together to increase the linear intervals of the sensors.

Finally, as an important part of the sensor, the packaging also ensured the service life and stability of the sensors. Processing the packaging commercially instead of using laboratory methods would improve the performance of the sensors to a certain extent, which will be the future direction of our sensor preparation.

## 4. Conclusions

In order to guarantee the safety of humans in human–robot interaction, this study designed a flexible tactile sensor system that could enhance the force perception of collaborative robots. The flexible sensor units that were made by adsorbing graphene onto a sponge could accurately convert contact surface pressure into numerical signals. Ecoflex was selected as the substrate material for the sensing array so as to make the sensors softer. A 3D printing mold was used to prepare the flexible shells of the sensors, which made the positioning of each part within the sensors more accurate and ensured the unity of the sensing array. This study also designed an integrated data acquisition circuit to collect data in real time and visualize the force data from the flexible tactile sensing units. The experiments showed that the flexible tactile sensing units could detect forces sensitively within the range of 0–5 N. The stability and reproducibility of the flexible tactile sensors were verified using experiments. All of the experimental results showed that the prepared electronic skin and the safety strategies could provide safe detection capabilities for robots during human–robot interaction. The softness of the electronic skin enabled it to adapt to the surfaces of various robots and provide them with sensing capabilities for external forces. The results of this study could be applied to relevant industrial scenarios, which could be of great significance for promoting the scientific development of robot safety control and the efficiency of production activities. In addition, collaborative robots are gradually being applied to home care. The application of electronic skins could make the work of collaborative robots more flexible and safer, which could open new opportunities for the future development of the home care industry.

## Figures and Tables

**Figure 1 micromachines-13-01344-f001:**
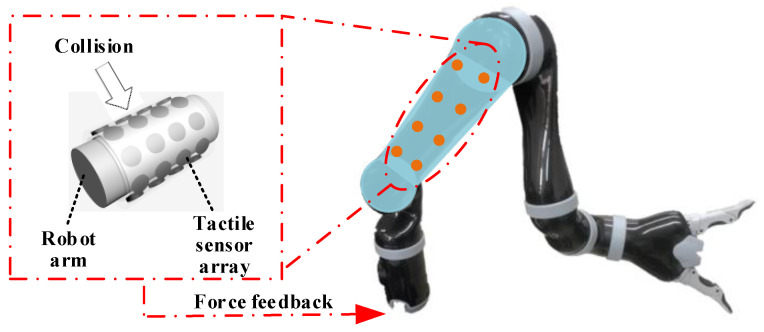
The strategy to obtain an energy-efficient trajectory.

**Figure 2 micromachines-13-01344-f002:**
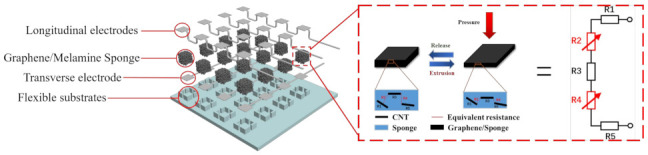
The sensing array of the 3D-printed substrate.

**Figure 3 micromachines-13-01344-f003:**
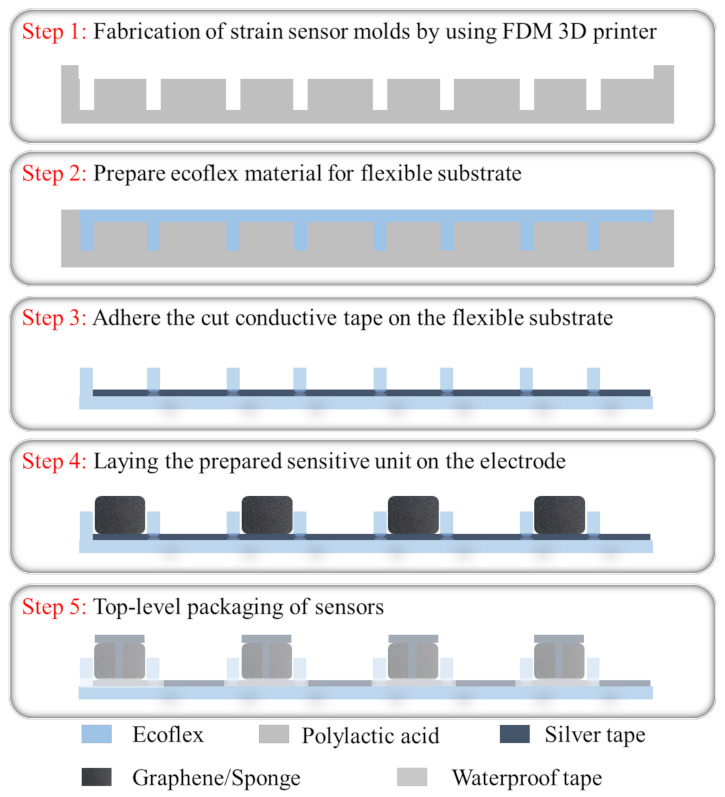
The fabrication process of the flexible sensing array.

**Figure 4 micromachines-13-01344-f004:**
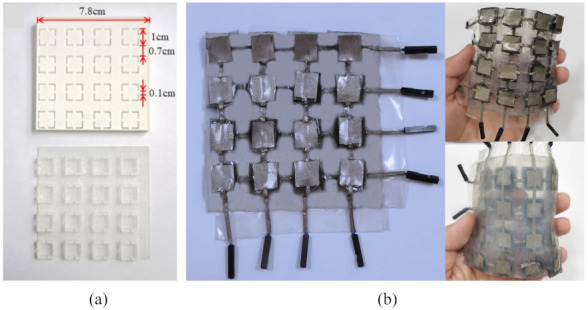
(**a**) A schematic diagram of the mold structure; (**b**) a photograph of the integrated flexible tactile sensor array.

**Figure 5 micromachines-13-01344-f005:**
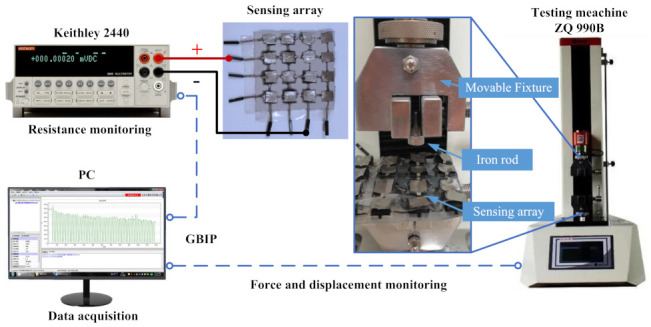
The configuration of the calibration and stability testing platform.

**Figure 6 micromachines-13-01344-f006:**
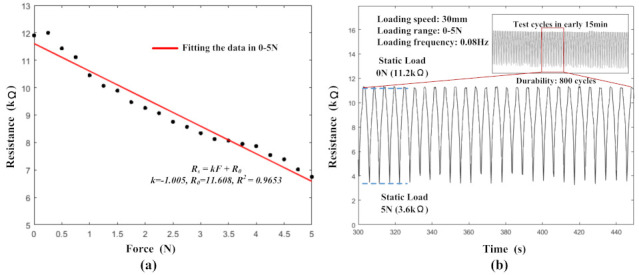
The calibration and stability testing of the tactile sensor units: (**a**) the calibration of the tactile sensor units; (**b**) the stability and reproducibility testing of the tactile sensor units.

**Figure 7 micromachines-13-01344-f007:**
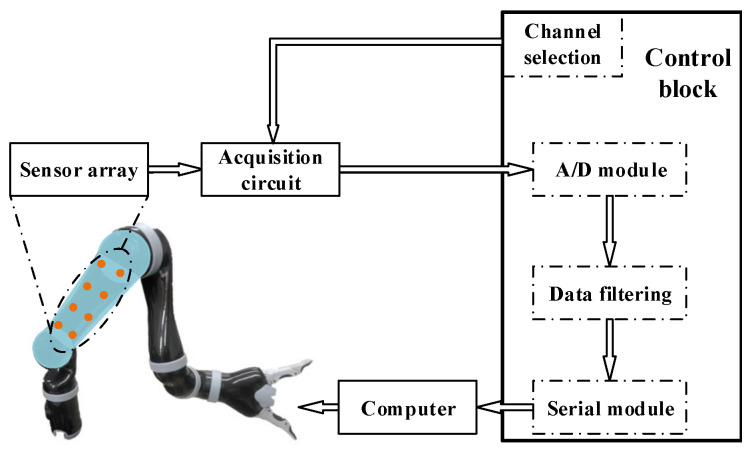
A schematic of the tactile acquisition system.

**Figure 8 micromachines-13-01344-f008:**
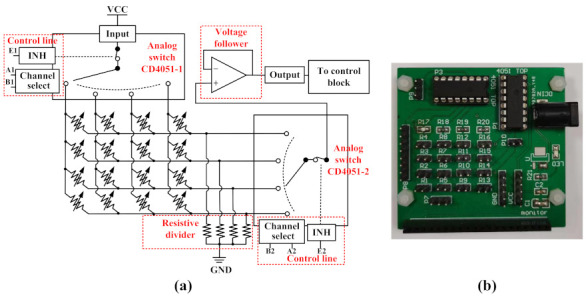
(**a**) A schematic of the tactile acquisition circuit; (**b**) a photograph of the acquisition circuit.

**Figure 9 micromachines-13-01344-f009:**
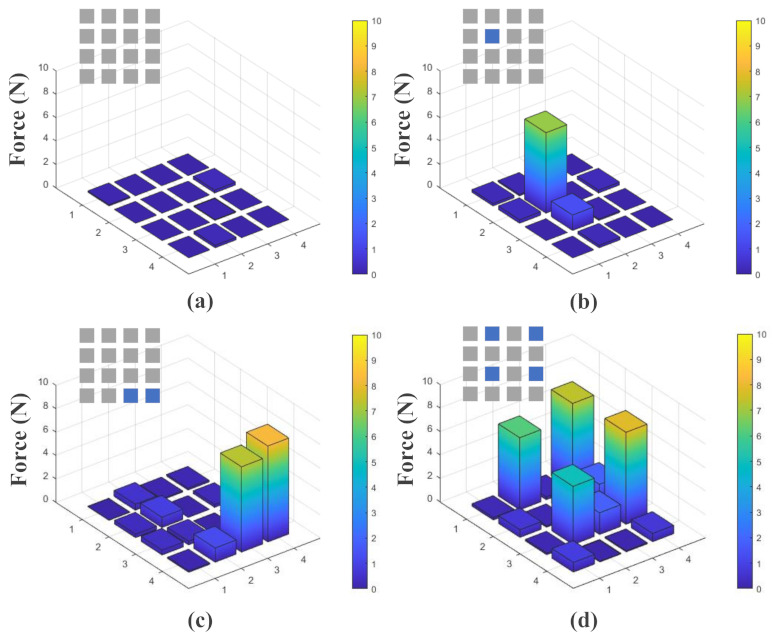
The distribution of the tactile data from the flexible tactile sensor array during finger touch tests under different conditions: (**a**) no sensing units being touched; (**b**) one sensing unit being touched; (**c**) two sensing units being touched; (**d**) four sensing units being touched.

**Figure 10 micromachines-13-01344-f010:**
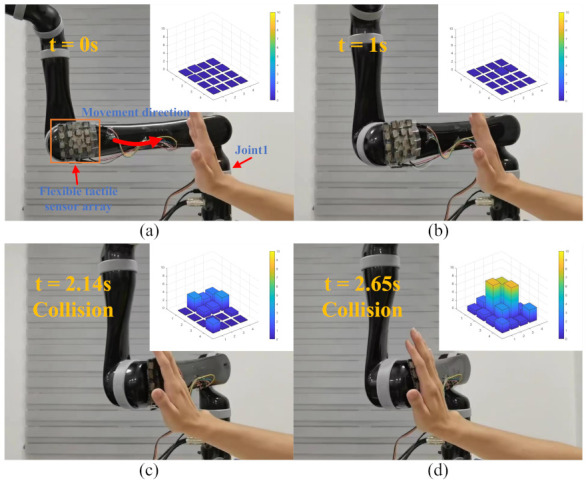
The safe human-robot interaction experiments. (**a**) the initial scene of experiments. (t = 0 s) (**b**) the robot moves without touching (t = 1 s) (**c**) the speed of the robot arm starts to decrease (t = 2.14 s) (**d**) the robot arm stops moving. (t = 2.65 s).

**Figure 11 micromachines-13-01344-f011:**
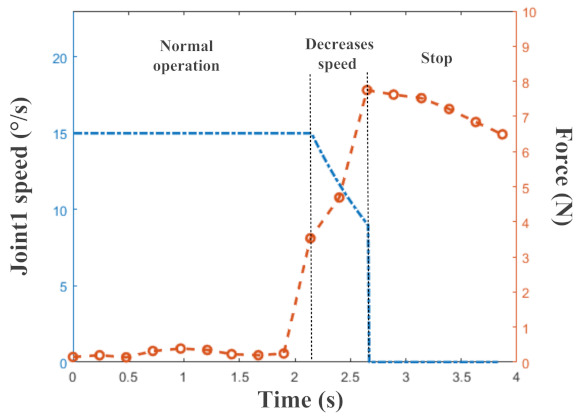
The speed of Joint 1 in the robot arm (blue curve) and the maximum force of the sensor array (red curve) during the human–robot interaction experiments.

**Figure 12 micromachines-13-01344-f012:**
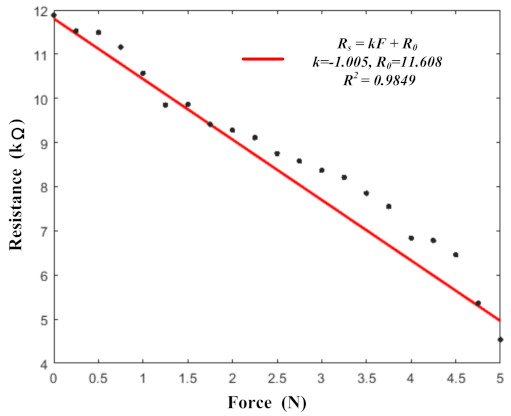
The fitting analysis of the benchmark sensing unit and the remaining 15 sensing units.

**Figure 13 micromachines-13-01344-f013:**
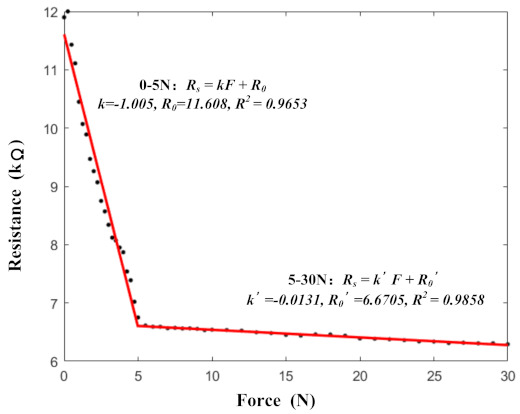
The two sensitivity intervals of the proposed sensors.

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
