# Peer review of "Implementation of a Sponge-Based Flexible Electronic Skin for Safe Human–Robot Interaction"

_micromachines, 2022, doi:10.3390/mi13081344_

Round 1

Reviewer 1 Report

In this manuscript, a flexible sensor is designed and made by adsorbing graphene into the sponge. The authors proposed a strategy based on the flexible sensor they made for safe human-robot interaction and tested it on the KINOVA robot arm. They have also done the stability and reproducibility experiments to show the sensor has good stability. And the interaction experiment they have done on the KINOVA shows the flexible sensor they made can play an important role in secure human-robot interaction.

 There are some problems should be noted:

1)      Many researches have been made on resistance array tactile sensors using foam as the sensing element. It’s my view that this article is not innovative enough.

2)      Will the sensing units affect each other when the sensor array detects pressure distribution? How to eliminate or reduce the effect?

3)      The linearity of the sensor mentioned in this article is not good enough. What's the reason? How to improve it?

4)      The static pressure distribution detection performance of the sensor is not mentioned in this article. Please add it into your article.

5)  This article only records the collision detection experiment, which is unable to completely show the performance characteristics of the sensor. Additional experiments are needed.

6)      As key sensitive element, the fabrication of sponge adsorbing graphene should be discussed in detail.

Reviewer 2 Report

This paper proposes a strategy for safe human-robot interaction and introduces the material, manufacturing technology, and design of a tactile sensor system. This strategy can help to protect the human body and the robot arm. The authors also discussed the material, manufacturing process, and design of the tactile sensor's tactile system. The finished work is similar to references 17 and 21. Because the manuscript does not reflect its own theoretical innovation, it is more appropriate for engineering application journals. Given the shortcomings of the article, the following suggestions are made:

(1) The introduction and related works should be written separately.

(2) Why are the positions of electronic skin stickers in Figures 1, 5 and 10 inconsistent?

(3) The equipment used is not well introduced.

(4) There is no Discussion part.

(5) Conclusion is insufficient and lacks prspects of potential application in the future.

(6) There are too few references, especially those really related to this study. Try quoting the references in the first five years, and reduce the citation of website links.

(7) Each sensor has unique characteristics; when combined, they will produce varying data fluctuations for the same force. How can this be resolved? For example, when the same force is applied to 5N, the output resistance of each sensor varies, so how should the follow-up work be handled?

(8) The experimental force is 0 to 5 N. Does the inertia force generated by the robot motion exceed this value? What would be the solution if it does? Is the data change consistent with the data change in the experimental stage?

(9) Whether there is a corresponding formula for resistance and pressure change, if so, it should be supplemented.

(10) The design part of the tactile system can be placed after the tension-compression test and repetitive cycle test of a single flexible tactile sensor.

Round 2

Reviewer 1 Report

This manuscript provides a sponge-based flexible sensor for safe human-robot interaction. Despite some modification was done in the revised version, the introduction part is still not  sufficient. The paragraph of sensorless and visual detection should be deleted, and the contents about tactile sensors should be increased. What's more, the authors should point out the problems existing in tactile sensors, and provide a solution against the problems.  In addition, the use of flexible sensor proposed in the manuscript is limited for detecting contact and collision between human and robot. But, the range of 0-5N is obviously not suitable.  

Reviewer 2 Report

The questions raised by the author have been answered, and the article has been modified. The article puts forward its own idea on the realization of the skin of a flexible robot with safe human-computer interaction based on the sponge.

The author conducts an experimental discussion on this idea, and the article conception and related work are improved.

Author Response

Thank you very much for your comments and professional advice, those comments are all valuable and helpful for revising and improving our paper, and provide important guidance to our research.